# Assessment of Excess Mortality in Italy in 2020–2021 as a Function of Selected Macro-Factors

**DOI:** 10.3390/ijerph20042812

**Published:** 2023-02-05

**Authors:** Emiliano Ceccarelli, Giada Minelli, Viviana Egidi, Giovanna Jona Lasinio

**Affiliations:** 1Statistical Service, Istituto Superiore di Sanità, 00161 Rome, Italy; 2Department of Statistical Sciences, La Sapienza University, 00185 Rome, Italy

**Keywords:** COVID-19, excess death, labour market areas, functional data, macrofactors, correlation, clustering model, regression model

## Abstract

Background: Excess mortality (EM) can reliably capture the impact of a pandemic, this study aims at assessing the numerous factors associated with EM during the COVID-19 pandemic in Italy. Methods: Mortality records (ISTAT 2015–2021) aggregated in the 610 Italian Labour Market Areas (LMAs) were used to obtain the EM P-scores to associate EM with socioeconomic variables. A two-step analysis was implemented: (1) Functional representation of EM and clustering. (2) Distinct functional regression by cluster. Results: The LMAs are divided into four clusters: 1 low EM; 2 moderate EM; 3 high EM; and 4 high EM-first wave. Low-Income showed a negative association with EM clusters 1 and 4. Population density and percentage of over 70 did not seem to affect EM significantly. Bed availability positively associates with EM during the first wave. The employment rate positively associates with EM during the first two waves, becoming negatively associated when the vaccination campaign began. Conclusions: The clustering shows diverse behaviours by geography and time, the impact of socioeconomic characteristics, and local governments and health services’ responses. The LMAs allow to draw a clear picture of local characteristics associated with the spread of the virus. The employment rate trend confirmed that essential workers were at risk, especially during the first wave.

## 1. Introduction

Over the last two years, Italy has gone through several waves of the COVID-19 disease, which brought about dramatic consequences for health [1]. According to the Italian National COVID-19 Surveillance System [2], 2,169,116 cases were recorded in 2020, and 4,237,257 were seen in 2021, with a total of 136,914 deaths. The impact of the COVID-19 pandemic on mortality is not complete when we consider only COVID-19 deaths and cases as reported by the countries’ surveillance systems. The total effects of a pandemic can be most reliably captured by the concept of *excess mortality* (EM) [3]. The measure represents *the difference between the number of deaths (from any cause) that occurred during the pandemic and the number of deaths that would have occurred in the absence of the pandemic*. Several proposals are found in the literature: model-based estimates of the deaths are commonly used at the national and regional scales [3,4,5]. Ceccarelli et al. (2022) compares several of them at county level [6]. EM can indicate the epidemic’s overall impact by highlighting its burden. That enables to prevent the underestimation of COVID-19 deaths impact (especially during the initial period of the pandemic) and allows deaths indirectly related to the disease, such as those due to delayed or missed treatment in the overburdened health care system, to be registered. EM is an essential metric for tracking the impact of a pandemic, both within and across countries. Analyses using EM have significant potential to aid in understanding which factors are most closely associated with the most severe outcomes during health crises. Many studies found that excess COVID-19 deaths represent only a portion of the EM observed since the start of the pandemic [7,8,9,10], indicating that indirect excess deaths may also contribute to the overall mortality burden.

Positive EM showed different gender and age profiles in many countries, suggesting that male and elderly people tend to be disadvantaged [11,12], while in the US, increases in mortality seemed to affect working-age people [13]. Some studies report that excess deaths are concentrated in socioeconomically disadvantaged areas [14]. All of these studies are conducted at a macro-level, and this work is key to understanding the impact of a pandemic according to population distribution, contact occasions, and health care system effectiveness in a given territory.

The present contribution is not focused on the metrics to produce a correct estimate of EM but rather on investigating which factors, at a macro level, were associated with EM. For this purpose, EM trends in Italy during the first two years of the pandemic were studied according to the Labour Market Areas (LMAs) [15], regions defined on commuting-to-work data, such that the majority of the labour force of an area lives and works within the boundaries of the LMA [16]. LMAs have long been recognised as essential for assessing the effectiveness of local and national policy decisions. This spatial unit makes it possible to analyse more homogeneous groups from a social and economic point of view. It highlights the different weights that specific socioeconomic and demographic determinants may have in estimating the impact of the pandemic in terms of mortality, given the same national mitigation measures. Furthermore, LMAs allow us to account for one of the most relevant contagion risk factors, namely, working activity.

EM time series were treated as functional data. First, a functional clustering approach was considered (Section 2.3) to identify similarities among LMAs at a national level. Then, functional regression models were estimated, relating EM to several socioeconomic variables in the specific clusters (Section 3).

## 2. Methods

### 2.1. Data Sources

All-cause mortality data from January 2015 to December 2021, publicly provided by the Italian National Institute of Statistics (ISTAT), were used. The data are at the municipality level by sex and age classes (age class in years at the time of death: 0, 1–4, 5–9, *…*, 95–99, 100+). A set of socioeconomic variables were selected to study the possible association with EM in Italy. These data are publicly available online at the LMAs level. Variable selection was based on Bernstein et al. (2001) [17] and Cadum et al. (1999) [18]. Among the many variables suggested in the literature as good proxies for the socioeconomic environment, those providing independent information (low correlation between them) were selected [19]. The correlation matrix is reported in Appendix A. Low education rate, unemployment rate, housing overcrowding, and raw rate of deaths for chronic disease were discarded. The selected variables were:**Low-income:** Frequency of residents with annual incomes between EUR 0 and 10.000. The variable was chosen to represent the poverty level of each LMA. Characterising LMAs as disadvantaged economic areas (or not disadvantaged) may enable us to understand whether a poor environment can be associated with positive EM. Source: Italian Revenue Agency, year 2019.**Population density**. Number of residents per km${}^{2}$. The variable is widely recognised as correlated with infections. Source: ISTAT, year 2011.**Proportion of over 70**: Age is a high-risk factor at the individual level. At the macro level, it may help to understand whether an ageing population structure is also associated with positive EM. Source: ISTAT, year 2021.**Beds per capita**: The indicator includes beds for acute patients in the ordinary regime. Each value represents the number of beds per 100,000 inhabitants, including LMAs within a radius of 80 km from each site. That considers the catchment area of a large hospital hub to be wider than the LMA in which it is located. The variable is a proxy of the health system’s local preparedness for the epidemic emergency immediately before the pandemic outbreak. Rough data source: Ministry of Health, year 2019.**Employment rate**: The employment rate tentatively highlighted possible connections between positive EM and the intensity of working activity. Source: ISTAT, year 2019.

Overall mortality and variables were aggregated at the LMAs geo-administrative unit level (LMAs composition source: ISTAT, 1 January 2021 [20]).

### 2.2. EM Estimation

EM is customarily estimated as a difference from an estimated baseline. The *P-score* was computed as the difference between the average weekly number of deaths in the years 2015 to 2019 (baseline) and the number of weekly deaths in the years 2020–2021. Finally, the difference was divided by the baseline itself [21].
(1)$$P_{it}=\frac{d_{it}-b_{it}}{b_{it}}$$
where $P_{it}$ is the P-score for the *i*th LMA in week *t*, $d_{it}$ is the observed number of deaths in the *i*th LMA in week *t*, and $b_{i}$ is the baseline computed in the *i*th LMA in week *t*. The P-score allows for comparisons between territories regardless of their size. For example, if a territory had a P-score of 0.5 in a given week in 2020, that would mean that the death count for that week was 50% higher than the expected death count for that week. EM values, computed in the 105 weeks of the two years 2020-2021 for all the 610 Italian LMAs, were analysed. Among the many possible EM estimation procedures, our choice was drawn from the methodology used in the Seventh joint ISTAT-ISS report (published 2 March 2022 [22]), where 163,942 excess deaths were estimated for the years 2020 and 2021 (total deaths reported: 746,146 and 709,035 in 2020 and 2021, respectively) using the average number of deaths in 2015-2019 in the country (645,620 deaths) as the baseline.

The ISTAT-ISS estimate of trends in EM does not deviate much from that found in the recent WHO publication [23], where a semiparametric model, estimated using the gam function in the mgcv package with generalised cross-validation, was adopted for the baseline estimation. The WHO analysis shows that localised time trends in EM occurred in all countries, especially during the first wave but also (though less heterogeneously) in subsequent waves.

### 2.3. Two-Stage Statistical Analysis

A two-stage procedure was adopted, combining a functional clustering of EM time series [24] and a linear functional regression model [25] to study the relationships between EM in the identified clusters and the available covariates. In Figure 1 a pictorial description of the analysis is summarised.

The EM time series was transformed into a functional object using the b-spline base system, which is usually preferred when modelling non-periodic data  [26]. The order of polynomials defining the basis functions was imposed as equal to 4, while the most appropriate number of basis functions was studied within the choice of clustering model. In the clustering step, the Discriminative Functional Mixture Model [27] was chosen, as implemented in the R package funFEM (https://cran.r-project.org/web/packages/funFEM/funFEM.pdf, accessed on 1 February 2022). This step is based on the homonyms algorithm [24] and allows for a choice within a family of discriminative functional submodels generated by applying several constraints on parameters of the general model [24]. In this setting, the model choice involves defining the number of clusters, the number of basis functions to describe the functional data, and the type of submodel “that best” fits the data. The final choice was based on the BIC criterion, following Schwartz (1978) [28], which penalises the log-likelihood $\ell (\hat{\theta })$ as follows. For model M:(2)$$\mathrm{BIC}\left(M\right)=\ell \left(\hat{\theta }\right)-\frac{\xi (M)}{2}\log\left(n\right),$$
where ξ(M) is the number of free parameters of the model, presented in Appendix A, and *n* is the number of observations. In the second stage, the relationship between weekly EM and available information was examined by estimating regression models for the functional data, one for each cluster (the estimation was implemented in the R function bayes_fosr (https://www.rdocumentation.org/packages/refund/versions/0.1-24/topics/bayes_fosr accessed on 1 February 2022) of the R package refund (https://cran.r-project.org/web/packages/refund/refund.pdf accessed on 1 February 2022)). The study’s response variables are of the functional type, and the independent variables are scalar. The latter produces functional coefficients [25]. Eventually, to estimate the model parameters, the variational Bayesian (VB) method was chosen [29].

In the following, further technical details on the chosen approaches are presented.

#### 2.3.1. Clustering Model

In this section, the DFM model [24] is illustrated. Let $\left\{x_{1},\ldots ,x_{n}\right\}$ be independent realizations of an $L_{2}$-continuous stochastic process $X={\left\{X\left(t\right)\right\}}_{t\in [0,T]}$. As is customary, the functional expressions of the observed curves are not known, and access is limited only to the discrete observations $x_{ij}=x_{i}\left(t_{is}\right)$, at a finite set of ordered times $\left\{t_{is}:s=1,\ldots ,m_{i}\right\}$. The unknown curves are described as belonging to a finite-dimensional space spanned by a basis of functions $\left\{\psi_{1},\ldots ,\psi_{p}\right\}$. Hence
(3)$$X\left(t\right)=\sum_{j=1}^{p}\gamma_{j}\left(X_{j}\right)\psi_{j}\left(t\right),$$
where γ is a vector of *p* unknown coefficient. Estimation is carried out using least squares smoothing [24]. To cluster $\left\{x_{1},\ldots ,x_{n}\right\}$ let us assume that there exists a latent variable $Z=\left(Z_{1},\ldots ,Z_{K}\right)\in {\left\{0,1\right\}}^{K}$ indicating the group membership of *X*. In this setting, the clustering model aims at predicting the vector Z.

Let F[0,T] be a latent subspace of $L^{2}\left[0,T\right]$ assumed to be the most discriminative subspace for the *K* groups spanned by a basis of *d* basis functions ${\left\{\varphi_{j}\right\}}_{j=1,...,d}$ in $L^{2}\left[0,T\right]$, with d<K and d<p. The basis ${\left\{\varphi_{j}\right\}}_{j=1,...,d}$ is obtained from ${\left\{\psi \right\}}_{j=1,...,p}$ through a linear transformation $\varphi_{j}=\sum {=}_{l=1}^{p}u_{jl}\psi_{l}$ such that the matrix $U=(u_{jl})$ is orthogonal. Now let us denote with $\left\{\lambda_{1},\ldots ,\lambda_{n}\right\}$ the latent expansion coefficients of the curves $\left\{x_{1},\ldots ,x_{n}\right\}$ on the basis ${\left\{\varphi_{j}\right\}}_{j=1,...,d}$. These coefficients can be described as independent realisations of a latent random vector $\Lambda \in {ℜ}^{d}$. The coefficients for the two bases φ and ψ are also linked by the following equation
(4)Γ=UΛ+ε,
where $\varepsilon \in {ℜ}^{p}$ is a Gaussian random noise term with a zero mean and a covariance matrix Ξ. Given (Equation 4) and conditioning on Z,
(5)$$\Lambda_{|Z=k}∼N\left(\mu_{k},\Sigma_{k}\right),$$
can be written. Where $\mu_{k},\Sigma_{k}$ are the mean and the covariance matrix of the group *k*, respectively. Hence, the coefficients Γ have a Gaussian mixture as marginal distribution:(6)$$p\left(\gamma \right)=\sum_{k=1}^{K}\pi_{k}\phi \left(\gamma ;U\mu_{k},U^{t}\Sigma_{k}U+\Xi \right)$$
where ϕ is the standard Gaussian density function, and $\pi_{k}=P\left(Z=k\right)$ is the prior probability of the *k*-th group. Eventually, let us assume that the matrix Ξ verifies condition (3.6) in Bouveyron et al. (2015) work [24]. This notation implies that the variance of the data of the *k*th group is modelled by $\Sigma_{k}$, whereas parameter β models the variance of the noise outside of the functional subspace. The model is called DFM${}_{\left[\Sigma_{k},\beta \right]}$. By setting constraints on $\Sigma_{k}$ and β, the *family of DFM* models is obtained. All of the considered models are reported in Appendix A. Again, models are identified using the formalization that is illustrated in Table A1 of Appendix C [24].

#### 2.3.2. Functional Regression Model

According to Morris (2009) [25], given $Y_{i}\left(t_{j}\right)$, for i=1,…,N and $j=1,\ldots ,T_{i}$, a set of functional dependent variables, and $X_{ia}$ for a=1,…,p a set of scalar predictors. The generic functional response for the linear regression model is given by
(7)$$Y_{i}\left(t_{j}\right)=\sum_{a=1}^{p}X_{ia}B_{a}\left(t_{j}\right)+E_{i}\left(t_{j}\right),$$
where $B_{a}\left(t\right)$ is the functional-coefficient of the *a*th variable and represents the partial effect of the predictor $X_{a}$ on the response variable at position *t*. $E_{i}\left(t\right)$s are the curve-to-curve residual error deviations, which are assumed to be Gaussian independent variables with zero mean and covariance $S(t_{1},t_{2})$. Estimating the set of coefficients $B_{a}\left(t\right)$ is required to identify the functional linear regression model. Significance, as in a classical regression model, is assessed by testing whether if $B_{a}\left(t\right)$ is non-zero at each time t∈[0,T].

Following Goldsmith (2016) [29], a fully Bayesian model is implemented. The Gibbs Sampling method [30] is chosen to estimate the model parameters.

## 3. Results

### 3.1. Clustering

All available models in the DFM family (see Appendix A for a full list) were considered. Model choice involves the number of basis functions (between 10 and 15) and the number of clusters (between 2 and 10). Model evaluation through the BIC index is reported in Appendix A Using the notation of Section 2.3.1 model ${\mathrm{DFM}}_{\left[\alpha_{kj},\beta \right]}$ where $\Sigma_{k}=diag{\left(\alpha_{kj}\right)}_{j=1,\ldots ,d}$ with d=12 basis function was chosen. The clustering results are represented in Figure 2.

Cluster 1, in yellow in Figure 2, is characterised by an average EM equal to 0.13, the lowest value of the four clusters. The average curve shows two peaks corresponding to the first and second waves of the epidemic. After the second wave, the EM was nearly constant until the end of 2021. In this cluster, 174 LMAs were found, where 43.7% of the country’s total population lives. The LMAs in the cluster are distributed equally across Italy. This cluster was labelled the *Low EM cluster*. Cluster 2, in green in Figure 2, reports average EM equal to 0.27. Four peaks correspond with the four periods of high virus diffusion across Italy. A total of 169 LMAs belong to the cluster, 32.1% of the population, distributed mainly across north and south Italy, is present there. This cluster was labelled the *Moderate EM cluster*. Cluster 3, in blue in Figure 2, is characterised by an average EM equal to 0.33. Cluster 3 curve highlights how the EM is higher than the other clusters throughout 2021. Further, it tends to increase across the final months of 2021. The cluster’s EM begins to increase during the second epidemic wave (fall 2020). To this cluster, 153 LMAs belong, of which 107 (69.9%) are located in the south of Italy, and 10.8% of the population of Italy lives here. This cluster was labelled the *high EM cluster*. Cluster 4, in violet in Figure 2, is characterised by a large peak during the first epidemic wave (March and April 2020), where the cluster curve reaches a maximum value of 2.08, several observed deaths in a week that is twice the expected number. After the first peak, a negative one follows during the summer of 2020. That leads to the hypothesis of a harvesting effect in the LMAs in the cluster following the first epidemic wave [31]. The EM curve reaches another peak during the second wave, smaller than the first; from this point, it decreases constantly. The average excess is 0.36, the largest among the four clusters. Cluster 4 is the smallest, as only 114 LMAs belong to it, of which 75 are located in the north of Italy. Despite this, 12.2% of the Italian population lives here. From the map in Figure 2, it can be seen that the cluster is concentrated in the Lombardy region, where the death toll during the first epidemic wave was higher than in the other Italian regions. This cluster was labelled the *high EM-first wave cluster*. Appendix B reports the composition of clusters by geographical area in Figure A1. The distribution of the variables *population density* and *Proportion of over 70* is almost uniform among the clusters. The *employment rate* is larger in Cluster 4, followed by Cluster 2, Cluster 1, and Cluster 3. As expected, the reverse is seen for the variable *Low-income*. Large values for *beds per capita* are mostly found in the LMAs belonging to Cluster 4. Boxplots describing the distribution of the variables within clusters are in Figure A2 of Appendix B.

### 3.2. Functional Regression Models

A different functional regression model was estimated for each cluster. The dependent variable was the weekly EM, and the independent variables were the socioeconomic variables presented in Section 2.1. The same covariates were selected for all four models; the variables were standardised within each cluster. The estimates are functional coefficients representing curves evolving over the 105 weeks in the chosen time window. Its 95% confidence band was associated with each curve. Statistical significance in functional regression is defined by the exclusion of the confidence band at time point *t*. The interpretation of the estimated coefficients is similar to that of the usual simple regression setting. The results are presented in Figure 3.

The model performances by cluster are explored in the Appendix A with the use of root mean square error, Appendix A, and diagnostic plots, Appendix A. In the Appendix A it is also included a plot representing the estimated functional coefficients separated by cluster and covariate.

#### 3.2.1. Intercept

In all four models, the intercept curve corresponds to the average EM in the cluster. That describes the EM trend adjusted for the considered variables.

#### 3.2.2. Low-Income

The coefficient’s curve appears negatively associated with all areas in Cluster 1. In Cluster 4, the low-income coefficient’s curve reaches a significant negative peak during the first wave of the epidemic. It grows again during the summer of 2020 and then oscillates, becoming insignificant. The curves oscillate in the other two clusters and are not statistically significant for most weeks.

#### 3.2.3. Population Density

The coefficient’s curve shows a negative association with the EM or a non-significant one. It takes on negative values at high values of EM, i.e., the peaks during the first wave (Cluster 4) and the second wave (Clusters 3 and 4).

#### 3.2.4. Proportion of over 70

The curve corresponding to the proportion of citizens over 70 years behaves similarly to the population density curve. It takes on negative values in clusters 3 and 4 when EM is large.

#### 3.2.5. Beds Per Capita

The curves suggest coherence in behaviour for Clusters 3 and 4 with a positive value during the first wave and then an oscillation around the zero line. The two curves have overlapping confidence bands at several time windows, highlighting similarities between the two clusters. Clusters 1 and 2 show a lower association with the variable. All four curves start a weak growth at the end of 2021.

#### 3.2.6. Employment Rate

All four curves show a positive peak of the coefficient of the *employment rate* during the first epidemic wave, directly proportional to the size of the epidemic wave in the clusters. During the summer of 2020, a decrease is observed in all curves. A small coefficient curve growth appears during the second epidemic wave. All curves decreased, showing a negative trend as the vaccination campaign progressed. The reverse of this trend is observed during the last months of 2021. The curve of this coefficient is coherent with the EM trend.

## 4. Discussion

Italy was the first European country that confronted the COVID-19 pandemic. During the first wave, the geographical spread of the pandemic was heterogeneous. In the southern regions and the islands, the infection’s spread was limited; in central regions, it was on average higher than in the south, while in northern regions, the circulation of the virus was high [32]. The latter has been explained by observing that the northern regions have well-established trade links with China, high population density, high levels of internal travel, and significant industrial activity [33]. In the summer of 2020, the virus’ spread reduced, and by the end of September, increasing outbreaks affected the entire country [7] marking the beginning of the second wave. The first wave mainly affected the north of Italy, but the second involved the country from north to south. Around March 2021, a decline in the contagion occurred. In the year’s second half, a progressive upward shift characterised the epidemic curve, affecting the southern regions with greater intensity. The government took timely measures: at the beginning of March 2020, face-to-face teaching in schools at all levels and all universities were suspended throughout the country, along with the ban on travel for *unnecessary reasons*, the suspension of sports activities, demonstrations and events, and the closure of museums, cultural places, and sports centres [34]. On 11 March 2020, a new measure was promulgated to suspend everyday retail, commercial activities, food services (excluding grocery shops), religious celebrations, assemblies of people in public, and open access to public places. All of the above measures were partially withdrawn during the summer. In the fall, when the number of cases began to increase again, new containment measures were adopted [35].

At the end of December 2020, the vaccination campaign began, initially involving people over 80 years of age and those defined as vulnerable. Later on, all age groups above 12 years old were involved. One year later, on 31 December 2021, 74.9% of the Italian population above 12 years old completed the vaccination cycle [36]. Some attention must be given to the definition of necessary activities characterising all governmental measures. All industrial and agricultural activities, food retails (supermarket and shops), and pharmacies were considered *necessary activities* and were allowed to stay open.

The trend of the impact of COVID-19 in terms of EM was, before the vaccination campaign, coherent with the epidemic’s trend, considering that the median time between diagnosis and death in Italy was 12 days, and most deaths (89%) occurred within 30 days from diagnosis [37]. Here a parallel between the EM trend and the epidemic’s dynamic is drawn to clarify the choice of working with the EM. So, even though in 2020–2021 EM and the raw number of COVID-19 deaths have a similar trend [22,38], it is demonstrated in a ISTAT seroprevalence study [39] that the COVID-19 prevalence in Italy is underestimated. This work is connected to Biggeri (2021) [40], and Section 2.2 to further stress the methodological choices.

EM is classified into four clusters in this work to capture the different time profiles of the epidemic spread. The association between the EM and LMA populations is described highlighting differences in terms of the chosen socioeconomic characteristics, along with possible differential responses by regional governments and health services (which are regional in Italy) that remain latent in the model. The choice of analysing geographic areas defined based on economic dimensions and social connections [15], such as LMAs, allows a local dimension more in line with the areal characteristics that influence the spread of the virus, as it is within these spatial units that the web of individual and family relationships develops, friendship and neighbourhood networks are established. Work and leisure time are organised [41].

This study uses variables updated to 2019, before the pandemic outbreak. They cannot be considered exhaustive of all the predictors influencing the excess. Some authors (Gollwitzer et al. (2020) [42] or Porcher and Renault (2021) [43]), for example, show that factionalism can be a significant predictor of behaviour and, therefore, of contamination. It would certainly be interesting to analyse how political polarization may have influenced the behaviour of communities in an emergency. Furthermore, municipal elections in Italy are not co-occurring in all administrative sections; they also differ by type (administrative or political), with considerable variability in time. Hence, adding references to these occurrences may introduce too much variability and uncertainty.

Although the variables that describe the different territorial situations explain only a limited part of the observed differences in the EM temporal trend, associations with the observed clusters’ behaviours capture some relevant differences that influenced trends seen in the pandemic patterns in local contexts. It is then possible to distinguish between the two clusters with the highest EM. Clusters 3 and 4 have very different socio-demographic profiles: Cluster 3, mainly affected by the epidemic during the second wave and in 2021, had the highest share of the low-income population of all clusters, the lowest employment rate, and the lowest health system endowments. These characteristics link Cluster 3 to Cluster 1, which has a low EM rate but a general socioeconomic disadvantage than the national average. Cluster 4 brings together the wealthiest LMAs in the country, where the employment rate is the highest (particularly in heavy manufacturing systems, Figure A3 of Appendix B) and the most significant health endowments. The epidemic hit Cluster 4 areas hard during the first wave. Then, a significant reduction in the EM followed. EM settled at levels close to Cluster 3 in the second half of 2020. Eventually, it fell lower than the same cluster in 2021. These two profiles clearly illustrate two different characteristics of the spread of the virus in Italy. On the one hand, part of the country suffered the devastating initial wave because of the influence of the more significant opportunities for contagion linked to economic activities and work [44]. The same was subsequently able to react more effectively, thanks to improved health structures and, perhaps, a more cooperative attitude on the part of the population. On the other hand, the one linked to the worst economic conditions and poor sanitary facilities, which, despite being hit later and with less intensity, was unable to effectively stem the contagion and the risk of death, recording the most significant excess. Comparing Clusters 1 and 2 with low and moderate EM provides further indications. Here, the trends in EM are very similar but show levels for Cluster 1 that are much lower than Cluster 2. The socio-demographic profiles that characterise the two clusters differ both due to the better economic conditions and the higher labour participation of Cluster 2 (mainly in the industrial sectors, while in Cluster 1, systems without specialisation and non-manufacturing systems prevail) and due to the higher population density and the higher health endowment of Cluster 1, which includes many of the large tertiary cities of the country (for example Rome, Bologna, Florence, and Cagliari). The COVID-19 timeline in Italy in 2020 and 2021 is shown in the Appendix A.

In Dorrucci et al. (2021) [7] and Gianicolo et al. (2021) [45] the EM is estimated with age class, showing a higher mortality risk for the elderly population. In this work, analysing group-based aggregate effects, we cannot draw conclusions at the individual level. The proportion of the elderly’s coefficient is mostly not significant, except for two clusters with lower EM, clusters 1 and 2, during the first wave, showing a positive association with EM. Beginning with the second wave, the largest deaths were recorded in areas with fewer elderly people. This finding prompts several suggestions that are currently not verifiable due to the lack of mortality data by cause of death.

In Biggeri (2021) [40] the relevance and complexity of COVID-19 data modelling are deeply discussed. The not-too-technical approach adopted allows both technicians and the general audience to ponder this very relevant issue. Data quality, model sensitivity, and distributional assumptions may always be questionable choices made by particular researchers. However, working with EM may be safer than many other approaches when data quality is not fully guaranteed. For example, when the health system is overwhelmed, direct measures are often not very accurate in the initial phases of an epidemic. The direct modelling of quantities such as the presumed COVID-19 deaths or cases is more affected by data quality than the EM. EM estimates require the definition and a possibly model-based, estimated baseline, and many approaches are found in the literature [4,5], always analysing average to large territorial units. In this paper, small spatial units are analysed (LMAs with surface areas ranging from 391 km${}^{2}$ in Cluster 4 to 571 km${}^{2}$ in Cluster 1; in terms of population size, the smallest LMA is found in Cluster 1 with 3156 inhabitants, the largest in Cluster 2, with almost 3.5 million inhabitants), introducing a strong methodological challenge in terms of estimation accuracy. Currently, the adopted methodology, through the computation of P-scores, ensures territorial comparability among units and with the Italian national statistical institutions (ISTAT-ISS [22]) working on the same data on the same time windows.

### Limitations and Future Developments

Data availability can be an issue in the current study. Municipal data are required to build information at the LMA level, and they are not always available. Future developments will include integrating a small-area approach to this type of study to allow the adoption of a fully model-based approach to the baseline estimation in small LMAs.

## 5. Conclusions

In this paper, excess mortality (EM) caused by the Covid-19 pandemic is classified into four clusters to capture the different temporal profiles of the epidemic spread. The association between EM and Labour Market Areas populations is described by highlighting differences in terms of selected socioeconomic characteristics, along with possible differential responses by regional governments and health services (which in Italy are regional) that remain latent in the model. Although the variables describing different territorial situations explain only a limited part of the observed differences in EM time trends, the associations with the cluster’s behaviors capture some relevant differences that influenced the observed trends in pandemic patterns in local contexts. The profiles identified in the clusters clearly illustrate the different characteristics of the virus spread in Italy.

## Figures and Tables

**Figure 1 ijerph-20-02812-f001:**
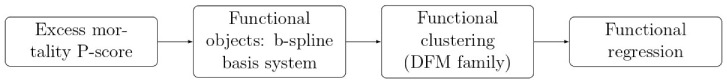
Statistical analysis workflow.

**Figure 2 ijerph-20-02812-f002:**
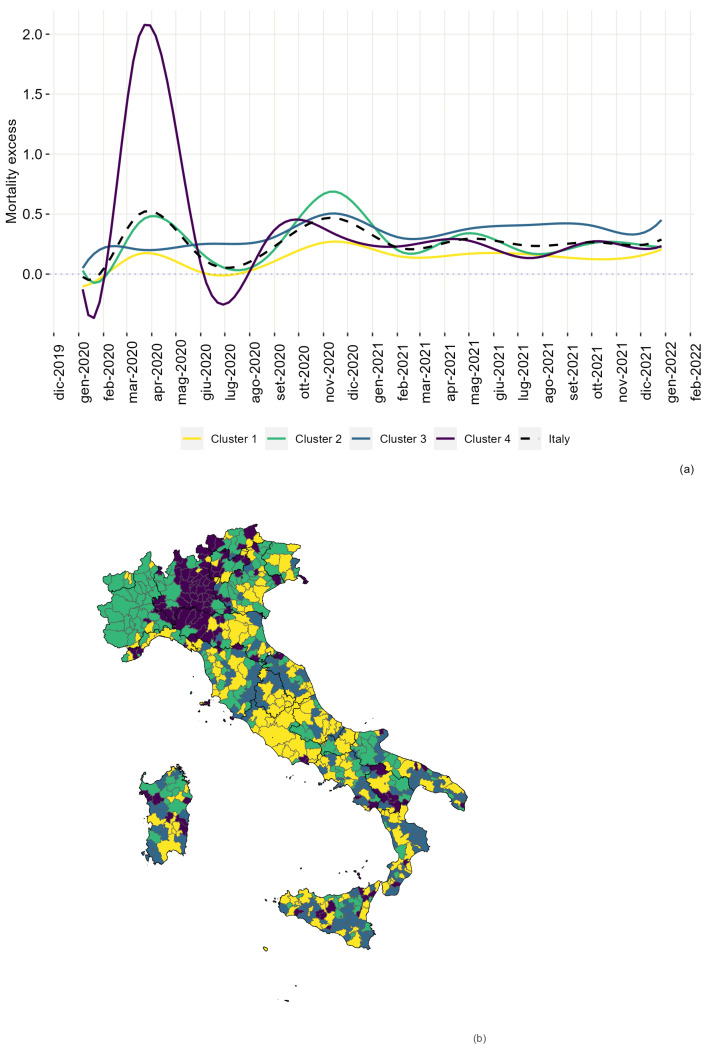
Average EM by cluster (**a**) and cluster distribution map (**b**) for the ${\mathrm{DFM}}_{\left[\alpha_{kj},\beta \right]}$ model with 12 bspline basis function, and four cluster. Cluster 1 curve is given in yellow, Cluster 2 in green, Cluster 3 in blue, Cluster 4 in violet.

**Figure 3 ijerph-20-02812-f003:**
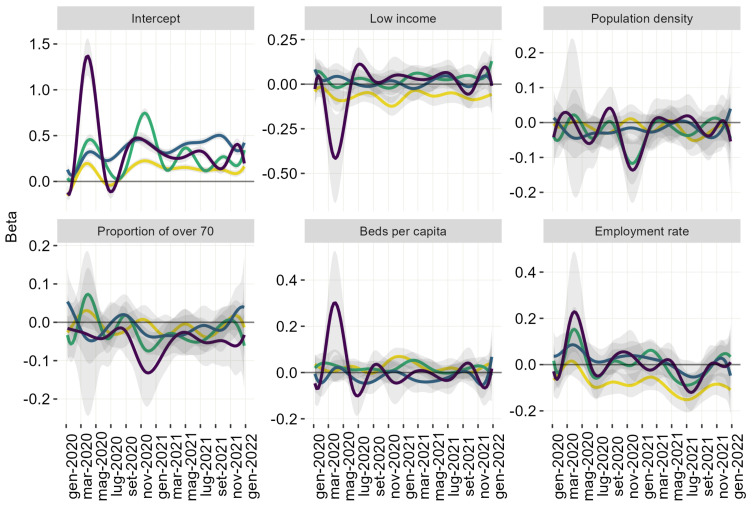
Estimated functional coefficients. The colour of the curves follows the colours of the clusters, namely, yellow—Cluster 1, green—Cluster 2, blue—Cluster 3, and violet—Cluster 4.

## Data Availability

Mortality data available at ISTAT (https://www.istat.it/it/archivio/240401 accessed on 14 June 2022). LMAs composition available at ISTAT (https://www.istat.it/it/archivio/252261 accessed on 18 August 2022). Population density available at ISTAT (https://www.istat.it/it/archivio/156224 accessed on 14 June 2022). Proportion of over 70 available at demo-ISTAT (https://demo.istat.it/ accessed on 14 June 2022). Low-Income available at Italian department of Finance, Economy Ministry (https://www.bing.com/ck/a?!&&p=6f011666092f91aeJmltdHM9MTY1OTM2MjA1MSZpZ3VpZD1mN2QyOWE2Ni1hODk3LTQ4NDEtOTEwOS0xZDI3YzI1MGU5NGYmaW5zaWQ9NTE2Mg&ptn=3&hsh=3&fclid=6b68fcfa-11a1-11ed-be02-67ea65b4f350&u=a1aHR0cHM6Ly93d3cxLmZpbmFuemUuZ292Lml0L2ZpbmFuemUvcGFnaW5hX2RpY2hpYXJhemlvbmkvcHVibGljL2RpY2hpYXJhemlvbmkucGhw&ntb=1, accessed on 14 June 2022). Employment rate available at ISTAT (https://www.istat.it/it/archivio/248606 accessed on 14 June 2022). Beds hospitals available at Ministry of Health (https://www.dati.salute.gov.it/dati/dettaglioDataset.jsp?menu=dati&idPag=18, accessed on 14 June 2022).

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
