# Peer review of "Assessment of Excess Mortality in Italy in 2020–2021 as a Function of Selected Macro-Factors"

_ijerph, 2023, doi:10.3390/ijerph20042812_

Round 1

Reviewer 1 Report

Please see my attached report.

Author Response

Major comment 1: The authors use the method of Goldsmith and Kitago [2016] which notably only uses variational methods for computational efficiency in model building and data exploration. For inference, Goldsmith and Kitago [2016] turn to a fully Bayesian model which better characterizes the variability in the estimation. The fully Bayesian version is implementable using the same bayes fosr function in R using bayes_fosr(formula, data = NULL, est.method = "Gibbs", cov.method = "FPCA", ...). This model should be estimated, at the very least, as a sensitivity analyses.

Response to major comment 1:  Thank you for pointing out the possible problems with the estimation method we used. We estimated again the functional coefficients, changing the estimation method (Gibbs Sampling) and reporting the results in the plot in figure 3. The new coefficients have the same trend as the previous ones but with a wider confidence interval.

Major comment 2: The authors argue, correctly, that excess mortality is a better indicator of the impact of COVID-19 than officially reported deaths. There is plenty of research to support this but it would make their case even stronger to show a similar analysis using the officially recorded deaths. That way the reader can clearly see the benefits gained from using excess mortality in such an analysis.

Response to major comment 2: Even though it could be interesting to work with the officially recorded COVID-19 deaths, this data is not yet available in Italy. Data with the specific cause of death is managed by the Italian National Institute of Statistics (ISTAT) and it is not publicly available. It needs to be requested in order to be used in research. In addition, the specific cause of death survey is subject to EU regulations that require the data to be released within 24 months of the reference year (https://www.istat.it/it/archivio/240401).

Another source of COVID-19 mortality data is the Italian National COVID-19 Surveillance System. Similarly as before, the data needed for our study is not publicly available at a municipality level (we recall that in order work with the LMAs, municipality-level data are needed) and it needs to be requested. These data probably does also contain too much noise due to various reasons: (1) the necessity of recording these deaths as soon as possible, (2) the changes in the definition of COVID-19 death throughout the two-year pandemic (https://www.epicentro.iss.it/), and (3) the underestimation of COVID-19 in Italy (https://www.istat.it/it/files//2020/08/ReportPrimiRisultatiIndagineSiero.pdf).

We added a sentence in the discussion to stress this last point (page 8 lines 264-266): “So, even though in 2020-2021 EM and the raw number of COVID-19 deaths have a similar trend, it is demonstrated in a ISTAT seroprevalence study that the COVID-19 prevalence in Italy is underestimated.

Major comment 3: Were the curves registered first? Given the delay in the spread, particularly early on, registration may be useful. It would align the curves and allow a more direct comparison of their shapes. Perhaps this could be used in the first stage, prior to clustering. The advantage of curve registration is that the peaks and valleys will be aligned, potentially making differences easier to observe. The disadvantage is that the time domain is distorted. Thus I would not recommend it for the second stage of the analysis where the authors clearly want to be able to refer back to calendar time. But in the clustering phase, it may be advantageous.

Response to major comment 3: EM curves were constructed from the mortality data processed by ISTAT. Contrary to the data elaborated by the Italian National COVID-19 surveillance system, there is no possibility of these being misaligned, in fact, these are the death data officially recorded by the National Register of Resident Population, throughout the country (https://www.istat.it/it/files//2022/03/Report_ISS_ISTAT_2022_tab3.pdf). In this way, all EM curves created are simultaneous and they don’t need to be registered.

Major comment 4: Pages 9 and 10, lines 307 to 312, the authors veer close to making an ecological fallacy in their discussion of the effects among older populations by insinuating individual level inference from group-based, aggregate effects. Please re-frame this discussion to avoid the appearance of suggesting individual level inferential impacts which cannot be drawn from the data at hand.

Response to major comment 4: We recognize the need to rephrase these lines to avoid possible misinterpretation of this result. We have added a sentence to avoid this (page 9 lines 312-313).

Minor comment 1: The referencing to some of citations is a bit awkward, largely due to the citation convention whereby only the number is stated in text.

Response to minor comment 1: We modified these references as suggested (page 1 line 28; page 2 line 69; page 5 lines 132, 138, 145; page 8 line 267; page 9 lines 311, 319).

Minor comment 2: In the top panel of Figure 2, the month abbreviations should likely be in English.

Response to minor comment 2: We have corrected the labels of the x-axis of Figure 2 and Figure 3, changing the month abbreviations in English.

Minor comment 3: Please thoroughly read and the edit the paper for grammatical mistakes and comprehension (clearing up the citation issue will help with comprehension in a number of places: page 1, line 28, page 5 line 132, page 9 line 263, page 9 line 292, page 9 line 307, page 10 line 313, there may be others).

Response to minor comment 3: We corrected all the citations you mentioned and other minor grammatical mistakes.

Minor comment 4: In the abstract, the authors use the term “correlate” several times. It would be more appropriate to refer to these as associations since correlations are neither discussed nor presented.

Response to minor comment 4: We have replaced the term “correlation” with the term “association” in the abstract.

Minor comment 5: For the non-specialist, the authors should define statistical significance in Section 3.2 after discussing the band construction (lines 194/195 on page 7). A simple “Statistical significance in functional regression is defined by the exclusion of the confidence band at time point t” or something to that effect would do.

Response to minor comment 5: As suggested, we have added the reported sentence to define statistical significance (page 7 lines 193-194).

Minor comment 6: The Figure reference is undefined on page 3 line 116.

Response to minor comment 6: We corrected the mistake and defined the reference on page 3, lines 116.

Minor comment 7: The color scheme in the top graph of Figure 2 is difficult to read since they are all warm colors. Please vary the colors more broadly. Also consider varying the line types for black and white printing.

Response to minor comment 7: In order to facilitate reading and improve accessibility for people with color vision deficiency, we have modified the color scheme.

Minor comment 8: The bands on Figure 3 are hard to see and difficult to differentiate since there is so much overlap between clusters. It is consequently difficult to confirm the authors’ interpretation of the results.

Response to minor comment 8: To resolve this problem we have added a new figure in the supplementary materials (figure S3) that will make it easier for the reader to read the results.

Reviewer 2 Report

Many thanks for allowing me to read this very important study. I enjoyed the relationship between structural factors and outcomes in terms of COVID-19. I am wondering whether you could consider financial factors, e.g. amount spent to deal with the social effects of COVID-19 (An, Porcher, Tang, Maille-Lefranc, 2022; Haug et al. 2020). Would your results hold with cases and deaths instead of excess mortality ?

I believe also that polarization might an important factor explaining differences among Italian region. Gollwitzer et al. (2020) or Porcher and Renault (2021) show that partisanship can be an important predictor of behavior, and thus contamination. Could you control for the results at the last national election for example ?

Author Response

Point 1: I am wondering whether you could consider financial factors, e.g. amount spent to deal with the social effects of COVID-19 (An, Porcher, Tang, Maille-Lefranc, 2022; Haug et al. 2020)

Response to point 1: It is surely possible and interesting to add more financial covariates in the study, other than low income, but we face the problem that it is not easy to find data available at a municipality level (in order work with the LMAs, municipality-level data are needed). 

Point 2: Would your results hold with cases and deaths instead of excess mortality ?

Response to point 2: Even though it could be interesting to work with the officially recorded COVID-19 deaths, this data is not yet available in Italy. Data with the specific cause of death is managed by the Italian National Institute of Statistics (ISTAT) and it is not publicly available. It needs to be requested in order to be used in research. In addition, the specific cause of death survey is subject to EU regulations that require the data to be released within 24 months of the reference year (https://www.istat.it/it/archivio/240401).

Another source of COVID-19 mortality data, and official cases, is the Italian National COVID-19 Surveillance System. Similarly as before, the data needed for our study is not publicly available at a municipality level (we recall that in order work with the LMAs, municipality-level data are needed) and it needs to be requested. These data probably does also contain too much noise due to various reasons: (1) the necessity of recording iy as soon as possible, (2) the changes in the definition of COVID-19 death and case throughout the two-year pandemic (https://www.epicentro.iss.it/), and (3) the underestimation of COVID-19 in Italy (https://www.istat.it/it/files//2020/08/ReportPrimiRisultatiIndagineSiero.pdf).

We added a sentence in the discussion to stress this last point (page 8 lines 264-266): “So, even though in 2020-2021 EM and the raw number of COVID-19 deaths have a similar trend, it is demonstrated in a ISTAT seroprevalence study that the COVID-19 prevalence in Italy is underestimated.

Point 3: I believe also that polarization might an important factor explaining differences among Italian region. Gollwitzer et al. (2020) or Porcher and Renault (2021) show that partisanship can be an important predictor of behavior, and thus contamination. Could you control for the results at the last national election for example?

Response to point 3: Thank you for this comment. It would certainly be interesting to analyze how political polarization may have influenced communities' behavior regarding the emergency situation. However, municipal elections in Italy are not all coordinated at the same time of year, so there may be too much variability and uncertainty. 

Round 2

Reviewer 1 Report

The authors have adequately addressed my comments. As such, I have no additional commentary.

Author Response

We thank you for your comments and we hope you found our work interesting. 

Reviewer 2 Report

I suggest the authors to directly consider my comments in the paper as limitations. The authors did respond to my comments but did not modify the manuscript in consequence or refer to the key concepts suggested.

Author Response

We added two sentences to address your comments and our responses in the manuscript.

On page 9, line 279-287: "This study uses variables updated to 2019, before the pandemic outbreak. They cannot be considered exhaustive of all the predictors influencing the excess. Some authors (Gollwitzer et al. (2020) or Porcher and Renault (2021)), for example, show that factionalism can be a significant predictor of behavior and, therefore, of contamination. It would certainly be interesting to analyze how political polarization may have influenced the behavior of communities in an emergency. Furthermore, municipal elections in Italy are not co-occurring in all administrative sections; they also differ by type (administrative or political), with considerable variability in time. Hence, adding references to these occurrences may introduce too much variability and uncertainty."

On page 10, lines 347-350, we created a new subsection "4.1. Limitations and future developments" with the added sentence: "Data availability can be an issue in the current study. Municipal data are required to build information at the LMA level, and they are not always available."